# Clustering Improves Differentially Private Inference

## Abstract

Differentially private (DP) language model inference is an approach for generating private synthetic text. A sensitive input example is used to prompt an off-the-shelf large language model (LLM) to produce a similar example. Multiple examples can be aggregated together to formally satisfy the DP guarantee.

Prior work creates inference batches by sampling sensitive inputs uniformly at random. We show that uniform sampling degrades the quality of privately generated text, especially when the sensitive examples concern heterogeneous topics.

We remedy this problem by clustering the input data before selecting inference batches. Next, we observe that clustering also leads to more similar next-token predictions across inferences. We use this insight to introduce a new algorithm that aggregates next token statistics by privately computing medians instead of averages. This approach leverages the fact that the median has decreased local sensitivity when next token predictions are similar, allowing us to state a data-dependent and ex-post DP guarantee about the privacy properties of this algorithm. Finally, we demonstrate improvements in terms of representativeness metrics (e.g., MAUVE) as well as downstream task performance. We show that our method produces high-quality synthetic data, at significantly lower privacy cost, than a previous state-of-the-art method.

## 1 Introduction

One of the many applications for powerful generative AI models is the creation of synthetic data. A natural approach is to prompt a large language model (LLM) with a rewriting task and a representative example, asking it produce synthetic analogs that resemble the example. This approach is not privacy-preserving if the seed example contains sensitive information that could theoretically pass through into the synthetic outputs.

This limitation is especially problematic if preserving the privacy of the source data was the reason to generate synthetic data in the first place. Consider a data steward who has access to a collection of medical records. They must preserve the privacy of the patients who provided the records. At the same time, they would like to make a privacy-preserving synthetic version of the data public to improve machine learning methods for making diagnoses.

The literature on *differentially private (DP) inference* (Dwork & Feldman, 2018; Papernot et al., 2017; 2018; Wu et al., 2024a; Ginart et al., 2022; Majmudar et al., 2022; Duan et al., 2023; Flemings et al., 2024a;b) provides a means to generate synthetic data by prompting a pre-trained model, while ensuring formal privacy guarantees (Hong et al., 2023; Tang et al., 2024; Amin et al., 2024; Gao et al., 2025). At a high-level, DP inference methods work by prompting an off-the-shelf LLM for multiple responses, with each one seeded by a sensitive example belonging to a different user. These responses are then aggregated in some way that satisfies DP. Through this procedure, an aggregated response does not represent any single seed example, but is a noisy amalgamation of all the seed examples.

In this work, we study the quality of synthetic data produced in this manner. In particular, we are interested in how the *heterogeneity* of the seed batch affects the *representativeness* of synthetic data.

DP requires that an adversary cannot detect any single seed example by observing the aggregated response. Thus, if data is highly heterogeneous, this presents a problem; by design, the aggregated response will not be representative of any seed example. In contrast, if all seed examples are highly

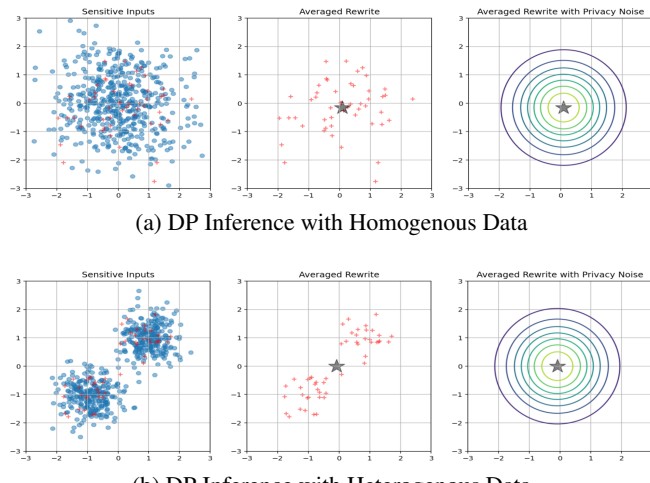

(a) DP Inference with Homogenous Data

(b) DP Inference with Heterogenous Data

Figure 1: A stylized depiction of synthetic data generation based on DP inference. *Left (a and b):* An input corpus sits in some embedding space, and is sampled uniformly at random (points in red) to form a batch. *Middle (a and b):* A depiction of an average rewrite for the batch. *Right (a and b):* A noise distribution centered around the average rewrite. In (b), the distinct semantic clusters found in the input dataset collapse.

self-similar, all responses will be similar, and the aggregated response can be representative of all the seed examples without violating the DP guarantee.

Armed with this observation we note that all state-of-the-art DP inference methods (Tang et al., 2024; Amin et al., 2024; Gao et al., 2025) batch seed examples uniformly at random, tending to generate heterogenous batches. We depart from this approach, demonstrating a practical technique for pre-clustering data while still preserving privacy. We use this clustering to assign similar examples to the same batch, creating more homogenous inputs for the DP inference algorithm.

Next, we propose a new algorithm for DP inference designed to make better aggregations when the LLM's predictions are aligned. We modify the algorithm of Amin et al. (2024), which aggregates LLM responses on a per-token basis by averaging token logit scores across inferences. Our algorithm replaces this average with a median. It is well-known in the DP literature that a median operation has local sensitivity that depends on how well-concentrated its inputs are. We prove a formal guarantee that holds in the data-dependent (Papernot et al., 2017; 2018) and ex-post (Ligett et al., 2017) differential privacy setting.

To the best of our knowledge, representativeness metrics like MAUVE (Pillutla et al., 2021) have not been previously evaluated for data generated via DP inference. Only recently, Amin et al. (2024) demonstrated generating enough data to begin measuring similarity at a dataset level. However, they report only accuracy measures on downstream tasks. Indeed, we show that their method fails to produce representative data as measured by MAUVE.

We conduct experiments on a variety of datasets and report improvement on two metrics: MAUVE scores computed on the raw synthetic dataset and accuracy of a BERT model trained on synthetic data. Finally, we incorporate a number of other improvements to further the state-of-the-art MAUVE scores, demonstrating the effect of other design decisions such as prompts, pre-trained vs. instruction-tuned generators, and varying the number of examples used when prompting.

## 2 LIMITATIONS OF UNIFORM BATCHING

As previously discussed, DP inference takes many text inputs (a batch) and attempts to produce a single output representing an *aggregated rewrite* for all of the records in the batch. At this level of abstraction, we can recognize a problem. Batches are ordinarily drawn uniformly at random from the input corpus. Therefore, the aggregated rewrite targeted by the these algorithms will collapse any of the variation within the corpus. Consider the visualization in Figure 1, where we think of text data

| Privacy $\epsilon$ | Method | MAUVE | Accuracy |
|---|---|---|---|
| | Real data | $.872_{.018}$ | $.965_{.001}$ |
| $\epsilon = \infty$ | Baseline (Amin et al., 2024) | $.130_{.009}$ | $.892_{.015}$ |
| | Baseline++ (w/ pretrained model & prompt) | $.460_{.050}$ | $.898_{.022}$ |
| | + *non-private clustering* | $.650_{.021}$ | $.912_{.020}$ |

Table 1: **Clustering improves DP inference results at $\epsilon = \infty$ on Yelp100k using Gemma 2 2B.** We report mean and std of *MAUVE* scores against real data (5 seeds), as well *Accuracy* of a BERT model trained on synthetic data and evaluated on real data (3 seeds). While Amin et al. (2024) enables generation of large synthetic corpora with DP inference, *quantity* begets the question of *representativeness*. **Baseline demonstrates the limits of existing approaches even when privacy is not a concern.** First, we show direct improvements via switching to the pretrained checkpoint and incorporating multiple examples into the prompt (*Baseline++*). On top of these improvements, **cluster-informed batching leads to improvements in representativeness**. Here, clustering is performed non-privately by running $k$-means on private data, with $K = 500$.

sitting in some embedding space, and the average response as a simple average within the embedding space. The distinct semantic clusters in in Figure 1(b) collapse due to the averaging procedure.

## 2.1 Empirical demonstration

We can demonstrate this claim by evaluating the performance of a DP inference method on a metric that captures the representativeness of the data generated. While the algorithm of Amin et al. (2024) is known to produce data that performs well on downstream classification tasks, these results do not tell us whether the synthetic data distribution represents the initial corpus. For that, we use MAUVE (Pillutla et al., 2021), a generic comparison measure between text corpora.

In Table 1 we see that DP inference (c.f. *Baseline*) does not produce representative datasets, even when pushing the methods to their limit by selecting parameters that offer no formal privacy guarantee (the $\epsilon = \infty$ regime). We begin our investigation from an improved baseline (*Baseline++*) obtained by (1) switching from Gemma 2 2B IT to the PT checkpoint (and necessarily changing the prompt); and (2) adding more in-context examples; full details are in Section 6.1).[1] The individual effects of each of these improvements can be found in Appendix D.1.

Conceptually, we can remedy the problem of heterogenous batches by first clustering the data, and then constructing batches by uniformly sampling inputs from *within each cluster*. For example, one could alternate between selecting batches from each of the 2 clusters in Figure 1. In Table 1, we report the MAUVE score of an algorithm (*non-private clustering*) that aims to do just that. The algorithm computes embeddings of the input corpus and clusters them using $k$-means. Batches are then constructed by first assigning inputs to clusters and feeding inputs with the same cluster assignment to the algorithm of Amin et al. (2024). While this procedure significantly improves MAUVE, it does not satisfy the DP guarantee. In the remainder of the paper, we describe: (1) how to implement this idea in a privacy-preserving manner; and (2) a new DP inference algorithm that takes advantage of pre-clustered data.

## 3 Preliminaries and notation

Let $\mathcal{X}$ be the token vocabulary, *i.e.*, the set of all possible tokens. A *token sequence* is an element of $\mathcal{X}^*$, and a *logit vector* is an element of $\mathbb{R}^{\mathcal{X}}$ (one logit per token in the vocabulary). For brevity we define $\mathcal{Z} \equiv \mathbb{R}^{\mathcal{X}}$ to be the set of all logit vectors. If $\mathbf{z} \in \mathcal{Z}$ then $z_x \in \mathbb{R}$ denotes the component of $\mathbf{z}$ corresponding to token $x \in \mathcal{X}$. If $\mathbf{x}_1$ and $\mathbf{x}_2$ are token sequences then we write $\mathbf{x}_1 \mathbf{x}_2 \in \mathcal{X}^*$ to denote their concatenation. A *large language model (LLM)* is defined by a function logits : $\mathcal{X}^* \to \mathcal{Z}$ that

---

[1]We find that the pre-trained (PT) checkpoint generates text that more closely matches the style and structure of the prompt, since that is what pre-training encourages, while the instruction-tuned (IT) checkpoint adds stylistic flourishes — for example, we observed the IT checkpoint inserting emojis into AGNews headlines.

maps each token sequence to a logit vector. A *dataset* $D \subseteq \mathcal{X}^*$ is a subset of token sequences. A pair of sets are *neighbors* if their symmetric difference has size 1, *i.e.*, one set can be formed from the other by adding or subtracting a single element.

## 4 IMPROVED ALGORITHM FOR DP INFERENCE

Algorithm 1 is our method for generating private synthetic text. Given a dataset of sensitive seed texts, the algorithm first partitions the seeds into $m$ batches. For each batch, the algorithm generates a single synthetic example consisting of $n$ tokens. Each synthetic example $\mathbf{x}$ is generated one token at a time, by first initializing $\mathbf{x}$ to be the empty token sequence and then repeatedly executing the following procedure: (1) generate $\text{logits}(\mathbf{sx})$ for each seed $\mathbf{s}$ in the batch, and aggregate the logit vectors into a single logit vector $\bar{\mathbf{z}}$; (2) draw token $x$ from $\text{softmax}(\bar{\mathbf{z}}/\tau)$, the distribution that assigns probability proportional to $\exp(\bar{z}_y/\tau)$ to each token $y$; (3) append $x$ to $\mathbf{x}$.

---

**Algorithm 1** Generate private synthetic examples

---

**Given:** $\text{logits} : \mathcal{X}^* \to \mathcal{Z}$, temperature $\tau > 0$, maximum token sequence length $n > 0$, $\text{batch} : \mathcal{X}^* \to [m]$, $\text{aggregate} : 2^{\mathcal{Z}} \to \mathcal{Z}$.
**Input:** Dataset of sensitive seeds $D \subseteq \mathcal{X}^*$.
**Output:** Dataset of synthetic examples $X \subseteq \mathcal{X}^*$.

1: $X \leftarrow \emptyset$
2: **for** each $i = 1, \ldots, m$ **do**
3:     $S_i = \{\mathbf{s} \in D : \text{batch}(\mathbf{s}) = i\}$.
4:     $\mathbf{x}_{i,0} \leftarrow$ Empty token sequence
5:     **for** $t = 1, \ldots, n$ **do**
6:         $Z_{i,t} \leftarrow \{\text{logits}(\mathbf{sx}_{i,t-1}) : \mathbf{s} \in S\}$
7:         $\bar{\mathbf{z}}_{i,t} \leftarrow \text{aggregate}(Z_{i,t})$
8:         $x_{i,t} \sim \text{softmax}(\bar{\mathbf{z}}_{i,t}/\tau)$
9:         Append $x_{i,t}$ to $\mathbf{x}_{i,t-1}$ to form $\mathbf{x}_{i,t}$
10:    $X \leftarrow X \cup \{\mathbf{x}_{i,n}\}$
11: **return** $X$

---

Algorithm 1 is a generalization of conventional non-private LLM inference, as well as the DP inference method of Amin et al. (2024). The differences between the methods are in their implementations of the $\text{batch}()$ and $\text{aggregate}()$ subroutines, which are marked in blue in Algorithm 1. In conventional inference, $\text{batch}()$ assigns each seed to its own unique batch, and $\text{aggregate}()$ has no effect. In the method from Amin et al. (2024), $\text{batch}()$ assigns each seed to one of $m$ batches uniformly at random (typically $m$ is much smaller than the number of seeds), and $\text{aggregate}()$ is defined

$$\text{aggregate}(Z) = \frac{1}{|Z|} \sum_{\mathbf{z} \in Z} \text{clip}_c(\mathbf{z}) \tag{1}$$

where $\text{clip}_c(\mathbf{z})_i = \max\{-c, z_i - \max_j\{z_j\} + c\}$. In other words, $\text{aggregate}()$ shifts and clips each logit value so that it lies in the interval $[-c, c]$, and then averages the clipped logit vectors together. Clipping is key to proving a privacy guarantee, which is based on the observation that the token sampling procedure is equivalent to the exponential mechanism (McSherry & Talwar, 2007).

### 4.1 BATCHING BY CLUSTERING

Instead of assigning seeds to batches randomly, in this paper we explore the impact of grouping similar seeds together. We consider implementations of $\text{batch}()$ in Algorithm 1 that have the form

$$\text{batch}(\mathbf{s}) = (\text{cluster}(\mathbf{s}), r) \tag{2}$$

where $\text{cluster}()$ is a cluster assignment function, and $r$ is chosen uniformly at random from $[b]$. In other words, the seed is first assigned to a cluster, and then within that cluster it is randomly assigned to one of $b$ batches. The cluster assignment function is implemented using a sentence embedding model $\text{embed}()$, which maps a given input text into a fixed-dimensional embedding

| Clustering Method | # Clusters | # Clusters (size $\geq$ 100) | Privacy $\varepsilon$ | V-measure |
|---|---|---|---|---|
| $k$-means | 500 | 464 | $\infty$ | 1 |
| DP Clustering (Chang & Kamath, 2021) | 9 | 6 | 0.1 | 0.1 |
| DP Clustering (Liebenow et al., 2024) | 2 | 2 | 10 | 0.01 |
| Public Centers | 438 | 140 | 0 | 0.59 |
| Public Centers with Rebalancing | 100 | 100 | 0.1 | 0.58 |

Table 2: Comparing different clustering methods. # Clusters is the number of non-singleton clusters, # Clusters ($\geq$ 100 Samples) is the number of clusters with at least 100 samples. V-measure (Rosenberg & Hirschberg, 2007) is a metric to compare the quality of each clustering. Higher V-measure shows more similarity to ground truth ($k$-means with $k = 500$). The parameter $\varepsilon$ indicates the privacy cost, with higher values indicating higher privacy cost (see Section 5).

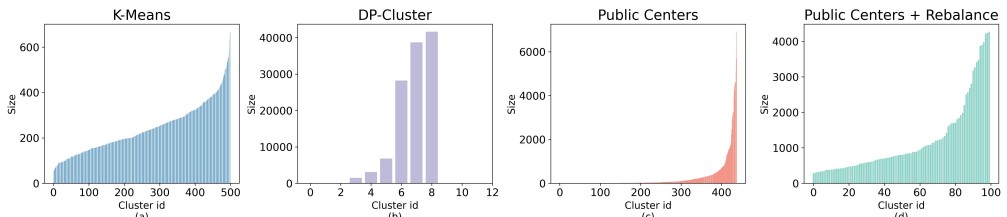

Figure 2: Cluster sizes of different clustering methods for AGNews dataset. a) $k$-means ($k = 500$) which is not private but gives the most balanced clusters. b) DP-clustering (Chang & Kamath, 2021) ($\varepsilon = 0.1$) which is private but most of the data is assigned to a few clusters. c) Clustering with public dataset (DBPedia, $k = 500$), which is private and has more valid clusters but still many clusters have only a few examples. d) Clustering with public centers and rebalancing (DBPedia, $k = 500$ and rebalancing to 100 clusters, $\varepsilon = 0.1$). This method does not have any small clusters.

space $\mathbb{R}^d$. A well-trained sentence embedding model will place similar texts closer together in this space. Given cluster centers $\mathbf{c}_1, \ldots, \mathbf{c}_k \in \mathbb{R}^d$ the cluster assignment function is defined as $\text{cluster}(\mathbf{s}) = \arg\min_{i \in [k]} \|\text{embed}(\mathbf{s}) - \mathbf{c}_i\|_2$. Thus the batching procedure is fully specified by describing how the cluster centers are selected. We consider the following three methods, each of which has different privacy implications.

**Differentially private centers.** We apply state-of-the-art DP clustering methods (Chang & Kamath, 2021; Liebenow et al., 2024) to the seed embeddings to discover the cluster centers. We observe that these methods often fail to find good cluster centers. Most DP clustering algorithms are designed for low-dimensional data, since the amount of privacy-preserving noise injected by the algorithms increases with the dimension, whereas sentence embeddings are typically high-dimensional.

Figure 2(b) shows one of the problems with DP clustering (Chang & Kamath, 2021). Even though the number of target cluster centers $k$ is set to 500, the algorithm only finds $< 10$ non-singleton centers, leading to highly imbalanced clusters.

**Public centers.** Given the limitations of privately clustering the seeds, we leverage high-quality public datasets instead. These datasets contain diverse examples, making them useful for clustering. We applied $k$-means clustering to the public data, and used the resulting centers to assign cluster labels to the seeds. Because selecting the cluster centers does not require examining any sensitive data, it does not incur any privacy cost. However, this approach introduces a new challenge: if the public data distribution differs significantly from the sensitive data distribution, the resulting clusters can become highly imbalanced, some with very few examples, and others disproportionately large. Very small clusters are often unusable, while large clusters may still contain heterogeneous data, reducing overall utility. Figure 2(c) illustrates the imbalance problem for public cluster centers.

**Public centers with rebalancing.** To address the cluster imbalance issue, we introduce two additional steps into the process of selecting cluster centers from public data. After obtaining public cluster centers, we compute a noisy count of the seeds assigned to each cluster; this step incurs only a small privacy cost ($\varepsilon \approx 0.1$). We then select the cluster centers with the $k'$ highest counts and re-assign the seeds using only these top-$k'$ centers. For example, if we had $k = 500$ centers originally, we may reduce them to $k' = 100$ centers after rebalancing. This refinement ensures more balanced clusters while preserving quality, as shown in Figure 2(d), improving both efficiency and utility.

## 4.2 MEDIAN AGGREGATION

In addition to improving synthetic data quality, making the batches more homogeneous allows us to modify Algorithm 1 to yield a tighter privacy analysis. Instead of aggregating the clipped logit vectors in a batch by taking their average, we compute their component-wise median:

$$\text{aggregate}(Z) = \text{median}(\{\text{clip}_c(\mathbf{z}) : \mathbf{z} \in Z\}), \tag{3}$$

where $\text{median}(Z)$ is the vector in which each component is the median value of the corresponding components of the vectors in $Z$. We discuss the usefulness of clipping in Appendix A.

Previous analyses of differentially private inference algorithms for synthetic data generation, such as in Amin et al. (2024) and Tang et al. (2024), were based on the *global* sensitivity of the mean, *i.e.*, on how much the mean of *any* set of logit vectors $Z$ can change when a vector is added or removed from $Z$. Our privacy analysis (in Section 5) is based on the *local* sensitivity of the median, *i.e.*, on how much the component-wise medians of the *actual* set of logit vectors $Z$ can change when a vector is added or removed from $Z$. To see why see the latter sensitivity can be much smaller than the former, note that if $Z$ contains at least 3 identical vectors, then the local sensitivity of $\text{median}(Z)$ is zero. When a batch of seeds texts are all similar to each other, then the logit vectors of their next-token distributions will also be similar. Furthermore, they will become increasingly similar as text generation proceeds, as the next-token distributions will become increasingly dependent on the generated text, and less dependent on the seed texts (see Figures 5 and 6 in Appendix D.4). We exploit this similarity to prove a stronger privacy guarantee than previous work (albeit one that is both data-dependent and output-dependent; see next section).

## 5 PRIVACY ANALYSIS

### 5.1 MEAN AGGREGATION PRIVACY GUARANTEE

In the standard definition of *approximate differential privacy* (Dwork et al., 2006), the upper bound is expressed in terms of privacy parameters $\varepsilon$ and $\delta$.

**Definition 1** (Approximate differential privacy). *Let $A : \mathcal{D} \to \mathcal{O}$ be an algorithm, $\varepsilon \geq 0$ and $\delta \in [0, 1]$. Algorithm $A$ satisfies $(\varepsilon, \delta)$-differential privacy if for all neighboring datasets $D, D' \in \mathcal{D}$ and $X \in \mathcal{O}$*

$$\Pr[A(D) = X] \leq \exp(\varepsilon) \cdot \Pr[A(D') = X] + \delta$$

We rely on an analysis due to Amin et al. (2024) to obtain a privacy guarantee for the version of Algorithm 1 that uses mean aggregation.

**Theorem 1** (Approximate differential privacy guarantee). *Algorithm 1 with* $\text{aggregate}()$ *set as Eq.* (1) *satisfies $(\varepsilon, \delta)$-differential privacy, where* $\varepsilon = O\left(n\left(\frac{c}{kb}\right)^2 + \frac{c}{kb}\sqrt{n \log \frac{1}{\delta}}\right)$.

### 5.2 MEDIAN AGGREGATION PRIVACY GUARANTEE

In contrast to Definition 1, and following more recent work (Papernot et al., 2017; Ligett et al., 2017; Papernot et al., 2018; Jordon et al., 2018; Chowdhury et al., 2020; Ginart et al., 2022; Duan et al., 2023; Flemings et al., 2024a), we also allow $\varepsilon$ to depend on both the input and output of the algorithm, which leads to a *data-dependent ex-post* guarantee.

**Definition 2** (Data-dependent ex-post differential privacy). *Let $A : \mathcal{D} \to \mathcal{O}$ be an algorithm. Let $\varepsilon : \mathcal{D} \times \mathcal{O} \to \mathbb{R}^{\geq 0}$. Algorithm $A$ satisfies $\varepsilon$-data-dependent ex-post differential privacy if for all neighboring datasets $D, D' \in \mathcal{D}$ and $X \in \mathcal{O}$*

$$\exp(-\varepsilon(D, X)) \cdot \Pr[A(D') = X] \leq \Pr[A(D) = X] \leq \exp(\varepsilon(D, X)) \cdot \Pr[A(D') = X]$$

Definition 2 reduces Definition 1 with $\delta = 0$ if we require the $\varepsilon$ function to be a constant function. In that special case, the privacy guarantee is a property of the algorithm itself, and holds for worst-case input and output.

**Towards a more-refined understanding of privacy risk.** Definition 2 offers the possibility of a privacy guarantee that is more refined than the worst case, reflecting the fact that certain inputs and outputs have lower privacy risk than others. This materializes in DP inference when we consider that *not all tokens pose the same privacy risk*: grammar and stop words such as "of", "the", ":", "and" – will be predicted *regardless of the private data*, hence we should not account the same privacy cost for releasing them as all other tokens. The same holds for tokens produced near the end of an example; these depend more on already-generated output and less on private data. Our data-dependent accounting indeed captures this behaviour: data-dependent $\varepsilon$'s drop significantly deeper into the generation (Figure 5).

In any case, the semantic meaning of the privacy guarantee is the same as in the standard definition of differential privacy: it quantifies the ability of an adversary to distinguish a small change in the input to the algorithm by only examining its output. Note that in Definition 2, the $\varepsilon$ value itself is not necessarily private (or 'sanitized'), but this feature is common in related work (Papernot et al., 2017; Jordon et al., 2018; Duan et al., 2023).

For the version of Algorithm 1 that uses median aggregation, we prove a novel privacy guarantee.

**Theorem 2** (Data-dependent ex-post differential privacy guarantee). *Algorithm 1 with* aggregate() *set as Eq. (3) satisfies $\varepsilon$-data-dependent ex-post differential privacy, where $\varepsilon(D, X) = \max_{i \in [m]} \sum_{t=1}^{n} \gamma(Z_{i,t}, x_{i,t})$, and the* per-token privacy cost *function $\gamma$ is defined in Appendix B.*

A formal definition of the function $\gamma$ in Theorem 2 is given in Appendix B, and here we offer some intuition for why it quantifies the privacy cost of Algorithm 1. For a set of logit vectors $Z$, let $Z^{(x)}$ be the values in the $x^{\text{th}}$ component of each of the vectors. These are the logit scores corresponding to token $x$. If we sort the logit scores in $Z^{(x)}$ in ascending order, then the median is the middle value, and the values that are adjacent to the median define what we call the *median gap*. When the adjacent values are far from the median, the median gap is large, and otherwise it is small. The size of the median gap determines the local sensitivity of the median, since adding or removing a value from $Z^{(x)}$ can cause the median to shift to one of the adjacent values. The function $\gamma(Z, x)$ is an increasing function of the median gap of $Z^{(x)}$, and so higher median gaps lead to higher privacy cost.

**Empirical privacy tests.** We supplement our theoretical analysis of our median aggregation algorithm with empirical privacy tests in Appendix D.5. This includes: a reconstruction test to see if our method memorizes private examples; and an empirical privacy audit to check for violations of our theoretical guarantee. In both cases, we do not find any evidence of privacy violations.

## 6 EXPERIMENTS

### 6.1 EXPERIMENT SETUP

**Models.** We use the pre-trained (PT) and instruction-tuned (IT) variants Gemma 2 2B models (Gemma Team, 2024) as the generator for all experiments. Note the variants necessitate using different prompts, which we give in E.4. All tasks use the same generic prompt template.

**Datasets.** Our algorithms utilize a *public dataset* in addition to the target *private dataset* we aim to synthesize. For private datasets in our experiments: we use *AGNews*, *Yelp*, and *NYT Topics*; all of which are equipped with a multi-class classification task. We use *DBPedia* as our sole public dataset for computing public clusters used in all experiments. We chose this dataset since it is based on Wikipedia, which (a) contains a wide variety of topics and therefore is a good candidate for universal clusters; and (b) reflects the kind of public data permissible for use in real deployments. For further details on all datasets used, see Appendix E.

**Evaluation.** We evaluate all methods on two metrics. **(1) BERT Accuracy:** we train a BERT model on synthetic data and report its final accuracy on a held-out set consisting of real data. To

| Dataset | Method | Privacy $\varepsilon$ | Clusters | MAUVE | Accuracy |
|---|---|---|---|---|---|
| AGNews | Real data | $\infty$ | - | $.872_{032}$ | $.938_{001}$ |
| | Mean Baseline (Amin et al., 2024) | 10 | 4 | $.156_{024}$ | $.704_{009}$ |
| | Mean Baseline++ | 10 | 4 | $.633_{022}$ | $.851_{015}$ |
| | Mean Clustered | $9.90 + 0.1$ | 60 | $.692_{029}$ | $.855_{012}$ |
| | Median Clustered | $2.40^* + 0.1$ | 60 | $.713_{027}$ | $.868_{002}$ |
| | Mean Baseline | 3 | 4 | $.141_{016}$ | $.701_{016}$ |
| | Mean Baseline++ | 3 | 4 | $.622_{024}$ | $.833_{006}$ |
| | Mean Clustered | $2.90 + 0.1$ | 60 | $.687_{034}$ | $.846_{002}$ |
| | Median Clustered | $1.22^* + 0.1$ | 60 | $.688_{046}$ | $.860_{004}$ |
| Yelp | Real data | $\infty$ | - | $.874_{012}$ | $.975_{000}$ |
| | Mean Baseline (Amin et al., 2024) | 10 | 2 | $.136_{014}$ | $.915_{006}$ |
| | Mean Baseline++ | 10 | 2 | $.415_{031}$ | $.899_{014}$ |
| | Mean Clustered | $9.90 + 0.1$ | 60 | $.449_{021}$ | $.906_{014}$ |
| | Median Clustered | $2.21^* + 0.1$ | 60 | $.460_{019}$ | $.912_{009}$ |
| | Mean Baseline | 3 | 2 | $.136_{012}$ | $.880_{014}$ |
| | Mean Baseline++ | 3 | 2 | $.391_{054}$ | $.907_{007}$ |
| | Mean Clustered | $2.90 + 0.1$ | 60 | $.436_{032}$ | $.904_{015}$ |
| | Median Clustered | $1.38^* + 0.1$ | 60 | $.451_{038}$ | $.904_{014}$ |
| NYT Topic | Real data | $\infty$ | - | $.863_{009}$ | $.919_{001}$ |
| | Mean Baseline (Amin et al., 2024) | 10 | 8 | $.151_{013}$ | $.668_{024}$ |
| | Mean Baseline++ | 10 | 8 | $.613_{045}$ | $.776_{008}$ |
| | Mean Clustered | $9.90 + 0.1$ | 80 | $.716_{038}$ | $.796_{001}$ |
| | Median Clustered | $5.04^* + 0.1$ | 80 | $.681_{043}$ | $.797_{001}$ |
| | Mean Baseline | 3 | 8 | $.155_{018}$ | $.668_{026}$ |
| | Mean Baseline++ | 3 | 8 | $.637_{055}$ | $.782_{007}$ |
| | Mean Clustered | $2.90 + 0.1$ | 80 | $.665_{017}$ | $.788_{002}$ |
| | Median Clustered | $1.72^* + 0.1$ | 80 | $.659_{053}$ | $.780_{006}$ |

Table 3: Performance of our methods compared to *Mean Baseline* (the algorithm of Amin et al. (2024)). We report the mean and std of MAUVE against real data (5 seeds) and downstream accuracy of a BERT model trained on the synthetic data (3 seeds). Our improved baseline (*Mean Baseline++*) shows sharp increases in MAUVE across all settings, as well as classification accuracy on AGNews and NYT Topic. On top of this stronger baseline, gains from clustering stack, and lead to consistent and direct improvements to MAUVE across all settings. For results employing clustering, we report the privacy cost of inference as well as the $\varepsilon = 0.1$ cost of cluster rebalancing. *Median Clustered* achieves better or comparable quality when compute-and-output-token-matched. (*) denotes an $\varepsilon$ value calculated using our ex-post data-dependent DP analysis.

compute BERT accuracy, we split the synthetic data into a synthetic train and validation set for model selection, and applying the best checkpoint on real data. **(2) MAUVE Score:** this metric which ranges from 0 to 1, measures the distributional similarity between the real and synthetic data. A higher score indicates better alignment, therefore, higher-quality synthetic data. We compute MAUVE with Gecko embeddings (Lee et al., 2024), using 1K samples from both sets.

**Baselines.** We implement the method of Amin et al. (2024) as a baseline, which only differs algorithmically in batching, and that the aggregated sampling logits is obtained via the mean only. Other DP inference synthetic data approaches in the literature (Tang et al., 2024; Gao et al., 2025) focus on generating few-shot examples for prompting, and have not demonstrated the ability to generate enough data ($\geq$2k examples) to compute MAUVE or finetune BERT. *Mean Baseline* is the setup described in Amin et al. (2024), using an IT model and prompt, 1 example per context, and within-label batching. *Mean Baseline++* uses 2 examples in-context and switches to a pre-trained checkpoint and prompt; ablations that decompose the effect of these changes can be found in D.1). Due to computational constraints, we tuned hyperparameters on AGNews and fixed them for the other datasets. For all experiments, we use a sampling temperature of 1.5. We use 64 parallel contexts for $\varepsilon = 10$ and 256 parallel contexts for $\varepsilon = 3$ experiments.

**Privacy budget.** We report results for two settings: $\varepsilon = 3$ and $\varepsilon = 10$. For mean aggregation, we set the approximate differential parameter $\delta = (\texttt{dataset\_size})^{-1.1}$. The privacy budget includes the total $\varepsilon$ used for both clustering and generation. For mean aggregation, we report unconditional $\varepsilon$.

For median aggregation, we match the number of tokens generated and batch size (thereby matching output quantity and compute requirements), and report the resultant data-dependent ex-post $\varepsilon$.

**Clustering and batching.** We start with 1000 clusters produced from DBPedia, and perform DP rebalancing as described in Section 4.1 to target 60 clusters for Yelp and AGNews, and 80 clusters for NYT Topics (10 clusters for each of the 8 labels). For implementation reasons, we subdivide each cluster into fixed-sized batches, rather than random-sized batches as in Eq. (2).

## 6.2 RESULTS

Table 3 summarizes our main results on all three datasets. We demonstrate that public-cluster-informed batching as a drop-in replacement for naive batching improves over the baseline – that is, simply adjusting the input batching to the algorithm and changing no other algorithmic details leads to significant improvements in representativeness. The same is the case for switching to the pretrained model, demonstrating stacking improvement. Furthermore, we show that our newly introduced median aggregation algorithm can achieve quality comparable or surpassing that of the mean algorithm, while admitting a tight, ex-post data-dependent DP analysis. We also remark that the privacy guarantee we give for the median algorithm is *maximum* over all batches. In Appendix D.4, we plot the distribution of per-batch privacy costs (many batches are <50% of the stated guarantee). Designing algorithms to take advantage of this property is an interesting avenue for future work.

## 7 RELATED WORK

**Differentially private synthetic data.** Prior work on generating synthetic data with differential privacy guarantees can be broadly categorized into three categories:

**A) Training-based** methods finetune language models on private data using differentially private stochastic gradient descent (Yue et al., 2023; Mattern et al., 2022; Carranza et al., 2024; Kurakin et al., 2023; Wang et al., 2024). After training, the model is used to generate synthetic data. More recent studies (Wu et al., 2024a; Tan et al., 2025; Tran & Xiong, 2024) leverage the abundance of public data by first finetuning the model on public datasets before applying differentially private finetuning on the private data.

**B) API-based** methods generate synthetic data using only model APIs (Xie et al., 2024; Yu et al., 2024; Wu et al., 2024b; Lin et al., 2024; 2025). They query the LLM with private examples and ask it to select the closest matching samples from a non-private dataset. They iteratively refine the output to ensure that it is similar to the private data.

**C) Inference-based** methods leverage private prediction (Dwork & Feldman, 2018), which ensures the privacy of model outputs (i.e., predictions). A widely used approach to achieve this is privacy amplification by subsampling and private aggregation (Nissim et al., 2007). When this methodology is applied to LLMs, the model generates the next token for each subset of private data, and the predictions are then privately aggregated to produce the final output (Hong et al., 2023; Amin et al., 2024; Tang et al., 2024; Gao et al., 2025).

**Differentially private clustering.** Early foundational work on differentially private clustering (Wang et al., 2015; Su et al., 2016; Feldman et al., 2009; Nissim et al., 2016) established strong theoretical bounds for private clustering. Subsequent works have improved the practical aspects, focusing on balancing utility, efficiency, and privacy. The most common approach (Balcan et al., 2017; Chaturvedi et al., 2020; Cohen-Addad et al., 2022) is to project the data into a lower dimension to reduce the additive error while preserving the relative distance, and then try to efficiently find good centers.

## 8 CONCLUSION

We have proposed a novel differentially private inference method for generating private synthetic data. Our method uses a clustering algorithm to group the input data into batches of similar examples, and leverages the resulting data homogeneity to generate high-quality synthetic data at significantly lower privacy cost.

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

## A   USEFULNESS OF CLIPPING FOR MEDIAN AGGREGATION

Given a logit vector $\mathbf{z}$, $\mathrm{clip}_c$ shifts each component by the same quantity so that $\max_j z_j$ becomes $c$. Any score below $-c$ is then clipped to $-c$. This operation bounds the range of possible values in $\mathbf{z}$ to $[-c, c]$ and does not complicate the privacy analysis since it operates locally on each $\mathbf{z}$. While medians are invariant to certain types of shifts, it is important to note that $\mathrm{clip}_c$ applies a different shift to each $\mathbf{z}$, while $\mathrm{median}$ aggregates across vectors. As a result, $\mathrm{clip}_c$ plays an important role in lowering the local sensitivity of the data. As a very simple example, consider a situation where all inferences are exactly aligned on the next-token distribution. Even if this alignment occurs, there is no reason that the *logit scores* would be aligned since the $\mathrm{softmax}$ operator is scale invariant. In contrast, the clipping operator forces alignment, while preserving the head of the distribution. In this example, the logit score for the most-likely next token will be mapped to $c$, driving the local sensitivity for this token down to zero.

## B   PROOF OF THEOREM 2

**Definition 3** (Left-median, median and right-median). *Let $Z \subseteq \mathcal{Z}$ be a set of logit vectors. Let* $\mathrm{left\text{-}median}(Z)$, $\mathrm{median}(Z)$, $\mathrm{right\text{-}median}(Z) \in \mathcal{Z}$ *be logit vectors, where the component of each vector corresponding to token $x$ is defined in terms of the multiset $Z^{(x)} = \{z_x : \mathbf{z} \in Z\} \subseteq \mathbb{R}$ as follows:*

- *If $|Z^{(x)}|$ is even, and $a$ and $b$ are the middle values in $Z^{(x)}$ (when all of the values are sorted), then* $\mathrm{left\text{-}median}(Z)_x = a$, $\mathrm{median}(Z)_x = (a + b)/2$ *and* $\mathrm{right\text{-}median}(Z)_x = c$.

- *If $|Z^{(x)}|$ is odd, and $a, b$ and $c$ are the middle values in $Z^{(x)}$ (when all of the values are sorted), then* $\mathrm{left\text{-}median}(Z)_x = a$, $\mathrm{median}(Z)_x = b$, $\mathrm{right\text{-}median}(Z)_x = c$.

*Note that since $Z^{(x)}$ is a multiset, it may contain repeated values, and therefore for any token $x$ it can happen that any of the consecutive values above are equal.*

The quantities in Definition 4 below depend on $\tau > 0$, but we have dropped this dependence from the notation to reduce clutter.

**Definition 4** (Per-token privacy cost function). *For any set of logit vectors $Z \subseteq \mathcal{Z}$ and token $x \in \mathcal{X}$ let*

$$\alpha(Z, x) = \exp((\bar{z}_x - \bar{z}_x^{\mathrm{right}})/\tau) \cdot \frac{\sum_y \exp(\bar{z}_y^{\mathrm{left}}/\tau)}{\sum_y \exp(\bar{z}_y/\tau)}$$

$$\beta(Z, x) = \exp((\bar{z}_x - \bar{z}_x^{\mathrm{left}})/\tau) \cdot \frac{\sum_y \exp(\bar{z}_y^{\mathrm{right}}/\tau)}{\sum_y \exp(\bar{z}_y/\tau)}$$

$$\gamma(Z, x) = \max\left\{\log \frac{1}{\alpha(Z, x)}, \log \beta(Z, x)\right\}$$

*where $\bar{\mathbf{z}}^{\mathrm{left}} = \mathrm{left\text{-}median}(Z)$, $\bar{\mathbf{z}} = \mathrm{median}(Z)$ and $\bar{\mathbf{z}}^{\mathrm{right}} = \mathrm{right\text{-}median}(Z)$.*

**Lemma 1.** *Let $Z, Z' \subseteq \mathcal{Z}$ be neighboring sets of logit vectors. For each token $x \in \mathcal{X}$ we have*

$$\mathrm{left\text{-}median}(Z)_x \leq \mathrm{median}(Z')_x \leq \mathrm{right\text{-}median}(Z)_x$$

*Proof.* Adding or removing a value from a multiset either leaves the median of the multiset unchanged, or shifts the median to the next higher or next lower value. $\square$

**Lemma 2.** *In Algorithm 1, suppose that batch $S_i$ is replaced by neighboring batch $S_i'$. For all $t \geq 1$, token $x \in \mathcal{X}$ and token sequence $\mathbf{x} \in \mathcal{X}^{t-1}$*

$$\alpha(Z_{i,t}, x) \leq \frac{\Pr[x_{i,t} = x \mid \mathbf{x}_{i,t-1} = \mathbf{x}]}{\Pr[x_{i,t}' = x \mid \mathbf{x}_{i,t-1}' = \mathbf{x}]} \leq \beta(Z_{i,t}, x)$$

*where $\mathbf{x}_{i,t}' = (x_{i,1}', \ldots, x_{i,t}')$ is the token sequence generated when processing batch $S_i'$.*

*Proof.* Let $\bar{\mathbf{z}} = \text{median}(Z_{i,t})$, $Z' = \{\text{clip}_c(\text{logits}(\mathbf{sx})) : \mathbf{s} \in S_i'\}$ and $\bar{\mathbf{z}}' = \text{median}(Z')$. We have

$$\Pr[x_{i,t} = x \mid \mathbf{x}_{i,t-1} = \mathbf{x}] = \frac{\exp(\bar{z}_x/\tau)}{\sum_y \exp(\bar{z}_y/\tau)}$$

$$= \frac{\exp(\bar{z}_x'/\tau)}{\sum_y \exp(\bar{z}_y/\tau)} \cdot \exp((\bar{z}_x - \bar{z}_x')/\tau)$$

$$= \frac{\exp(\bar{z}_x'/\tau)}{\sum_y \exp(\bar{z}_y'/\tau)} \cdot \exp((\bar{z}_x - \bar{z}_x')/\tau) \cdot \frac{\sum_y \exp(\bar{z}_y'/\tau)}{\sum_y \exp(\bar{z}_y/\tau)}$$

$$= \Pr[x_{i,t}' = x \mid \mathbf{x}_{i,t-1}' = \mathbf{x}] \cdot \exp((\bar{z}_x - \bar{z}_x')/\tau) \cdot \frac{\sum_y \exp(\bar{z}_y'/\tau)}{\sum_y \exp(\bar{z}_y/\tau)} \quad (4)$$

Continuing from above

$$\text{Eq. (4)} \geq \Pr[x_{i,t}' = x \mid \mathbf{x}_{i,t-1}' = \mathbf{x}] \cdot \exp((\bar{z}_x - \bar{z}_x^{\text{right}})/\tau) \cdot \frac{\sum_y \exp(\bar{z}_y^{\text{left}}/\tau)}{\sum_y \exp(\bar{z}_y/\tau)} \quad \because \text{Lemma 1}$$

$$= \Pr[x_{i,t}' = x \mid \mathbf{x}_{i,t-1}' = \mathbf{x}] \cdot \alpha(Z_{i,t}, x)$$

and

$$\text{Eq. (4)} \leq \Pr[x_{i,t}' = x \mid \mathbf{x}_{i,t-1}' = \mathbf{x}] \cdot \exp((\bar{z}_x - \bar{z}_x^{\text{left}})/\tau) \cdot \frac{\sum_y \exp(\bar{z}_y^{\text{right}}/\tau)}{\sum_y \exp(\bar{z}_y/\tau)} \quad \because \text{Lemma 1}$$

$$= \Pr[x_{i,t}' = x \mid \mathbf{x}_{i,t-1}' = \mathbf{x}] \cdot \beta(Z_{i,t}, x) \qquad \square$$

We are now ready to prove Theorem 2.

*Proof of Theorem 2.* Let $D, D' \in \mathcal{D}$ be neighboring datasets. For each seed, we can condition on a fixed value for the random integer $r$ selected in Eq. (2), since it is chosen independently of the dataset. Since the $\text{batch}()$ function assigns each seed to one batch, there exists a single batch that differs by one seed when Algorithm 1 is run on input dataset $D$ instead of $D'$. Let $S_i$ and $S_i'$ be these neighboring batches. Let $x_{i,1}, \ldots, x_{i,n}$ and $x_{i,1}', \ldots, x_{i,t}'$ be the sequences of tokens generated when processing $S_i$ and $S_i'$, respectively. For each $t \in [n]$ let $\mathbf{x}_{i,t} = (x_{i,1}, \ldots, x_{i,t})$ and $\mathbf{x}_{i,t}' = (x_{i,1}', \ldots, x_{i,n}')$ denote the first $t$ tokens of $\mathbf{x}_{i,n}$ and $\mathbf{x}_{i,n}'$, respectively. Also fix a token sequence $\mathbf{y}_n = (y_1, \ldots, y_n) \in \mathcal{X}^n$, and for each $t \in [n]$ let $\mathbf{y}_t = (y_1, \ldots, y_t)$ denote the first $t$ tokens of $\mathbf{y}_n$. We have

$$\frac{\Pr[\mathbf{x}_{i,n} = \mathbf{y}_n]}{\Pr[\mathbf{x}_{i,n}' = \mathbf{y}_n]} = \frac{\Pr[x_{i,1} = y_1]}{\Pr[x_{i,1}' = y_1]} \cdot \frac{\Pr[x_{i,2} = y_2 \mid \mathbf{x}_{i,1} = \mathbf{y}_1]}{\Pr[x_{i,2}' = y_2 \mid \mathbf{x}_{i,1}' = \mathbf{y}_1]} \cdots \frac{\Pr[x_{i,n} = y_n \mid \mathbf{x}_{i,n-1} = \mathbf{y}_{n-1}]}{\Pr[x_{i,n}' = y_n \mid \mathbf{x}_{i,n-1}' = \mathbf{y}_{n-1}]}$$

Taking logarithm of both sides and applying Lemma 2 we have

$$\sum_{t=1}^n \log \alpha(Z_{i,t}, x_{i,t}) \leq \log \frac{\Pr[\mathbf{x}_{i,n} = \mathbf{y}_n]}{\Pr[\mathbf{x}_{i,n}' = \mathbf{y}_n]} \leq \sum_{t=1}^n \log \beta(Z_{i,t}, x_{i,t})$$

which proves the theorem. $\qquad \square$

## C  RELATED WORK CONTINUED

**Clustering in training-based methods.**  As explained before training-based techniques involve fine-tuning a generative model on the private dataset using the DP-SGD algorithm. The fine-tuned model is then used to generate synthetic samples. A significant challenge with this approach is the potential for a distributional mismatch between the synthetic and real data.

Yu et al. (2024) first highlighted this distributional problem and proposed a solution utilizing a DP-histogram to obtain a more representative understanding of the contents of the private data. Their work demonstrated that filtering or resampling of the generated synthetic data, guided by the histogram information, could substantially improve the quality of the resulting dataset.

Subsequently, Tan et al. (2025) addressed the inherent limitations of a standalone DP-histogram, which may provide insufficient information about the private data. They proposed leveraging public datasets to improve the generation process. Specifically, their approach incorporates a sophisticated *universal topic model*, built on a large-scale public corpus (Wikipedia), alongside the dp fine-tuned generative model. The topic model is used to derive a differentially private topic histogram of the private data, which captures high-level distributional information. During the data generation phase, this topic histogram is employed to ensure that synthetic data is produced in proportion to the observed topic distribution in the private data.

**Comparison with our work.**  It is notable that while both Yu et al. (2024); Tan et al. (2025) employ histogram information of the private data and Tan et al. (2025) utilizes public data to inform its the generative model, their fundamental objective and application differ substantially from the methodology proposed in this paper.

The aforementioned works use clustering/topic modeling to guide the generative model (e.g., via resampling or proportional generation) to mitigate the distributional mismatch between the real and synthetic data. The topic model in Tan et al. (2025), for instance, captures semantic, high-level topical information that is crucial for improving the quality of training-based synthetic data generation.

First of all our proposed method is based on DP inference rather than DP-SGD training. Furthermore, our method utilizes clustering with public centroids for a distinct purpose: to group highly similar private data instances. Following this clustering step, the only information we retain is the cluster similarity of the private samples; the semantic topic or high-level content of the clusters remains unknown. This grouping of similar records is sufficient for our approach and represents a fundamental distinction from prior work that uses distributional information to help the generative process.

## D  EXPERIMENTS CONTINUED

### D.1  IMPROVED BASELINE

Our experiments begin by improving over the baseline in Amin et al. (2024). *Baseline++* includes two modifications: switching from an instruction-tuned model to a pre-trained model, and including more examples in-context. Both changes improve the representativeness of the generated data. We break down the individual effects of these modifications.

| Dataset | Privacy $\varepsilon$ | Checkpoint | # Examples | MAUVE |
|---|---|---|---|---|
| AGNews | $\infty$ | IT | 1 | $.152_{.007}$ |
|  |  | IT | 2 | $.218_{.012}$ |
|  |  | PT | 1 | $.559_{.023}$ |
|  |  | PT | 2 | $.640_{.046}$ |
| Yelp | $\infty$ | IT | 1 | $.130_{.009}$ |
|  |  | IT | 2 | $.203_{.024}$ |
|  |  | PT | 1 | $.385_{.015}$ |
|  |  | PT | 2 | $.460_{.050}$ |

Table 4: Effect on MAUVE from varying the number of in-context examples, and model checkpoint.

## D.2 Ablation on the number of clusters

We present additional results on the influence of the number of clusters, in the setting of Table 1. Note that Table 1 only reports the (Yelp, 500 clusters) result.

| Yelp | | Agnews | |
|---|---|---|---|
| **Clusters** | **MAUVE** | **Clusters** | **MAUVE** |
| 2 | $.460_{.050}$ | 1 | $.484_{.023}$ |
| 200 | $.663_{.031}$ | 4 | $.640_{.046}$ |
| 500 | $.650_{.021}$ | 500 | $.778_{.014}$ |

Table 5: MAUVE scores for different numbers of clusters on the Yelp and AGNews datasets.

## D.3 Influence of Embeddings

An important choice for our experiments is the embedding used when evaluating MAUVE. We re-run our experiments on the Yelp dataset switching from Gecko (Lee et al., 2024) to GPT2-large (Radford et al., 2019). The ordering of results in Table 3 is largely preserved.

| Dataset | Method | Privacy $\varepsilon$ | Gecko MAUVE | GPT2 MAUVE |
|---|---|---|---|---|
| | Real data | $\infty$ | $.874_{.012}$ | $.960_{.001}$ |
| | Mean Baseline (Amin et al., 2024) | 10 | $.136_{.014}$ | $.075_{.010}$ |
| | Mean Baseline++ | 10 | $.415_{.031}$ | $.555_{.054}$ |
| Yelp | Mean Clustered | $9.90 + 0.1$ | $.449_{.021}$ | $.580_{.024}$ |
| | Median Clustered | $2.21^* + 0.1$ | $.460_{.019}$ | $.601_{.023}$ |
| | Mean Baseline | 3 | $.136_{.012}$ | $.083_{.010}$ |
| | Mean Baseline++ | 3 | $.391_{.054}$ | $.568_{.025}$ |
| | Mean Clustered | $2.90 + 0.1$ | $.436_{.032}$ | $.623_{.029}$ |
| | Median Clustered | $1.38^* + 0.1$ | $.451_{.038}$ | $.581_{.029}$ |

Table 6: MAUVE scores on Yelp dataset using Gecko and GPT2 embeddings. An asterisk (*) next to an $\varepsilon$ value for the Median Clustered method indicates that it was calculated using our ex-post data-dependent DP analysis from Section 5.

## D.4 Median mechanism privacy cost

Unlike unconditional, worst-case $\varepsilon$ that admits *uniform* per-token privacy costs, the median mechanism's privacy cost differs batch-by-batch and also position-by-position. The figures below plot these privacy costs.

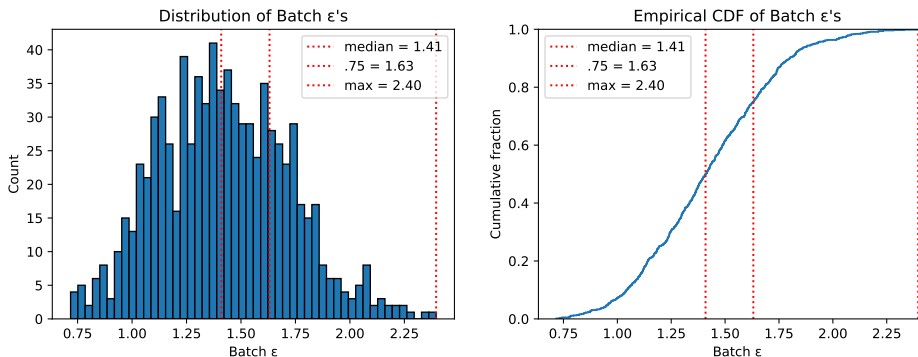

Figure 3: We plot the distribution of *per-batch $\varepsilon$ costs* of the median mechanism on AGNews. The maximum over all batches obtains $\varepsilon = 2.40$, which is the privacy guarantee we report in Table 3 via Theorem 2. Most batches have substantially smaller privacy cost.

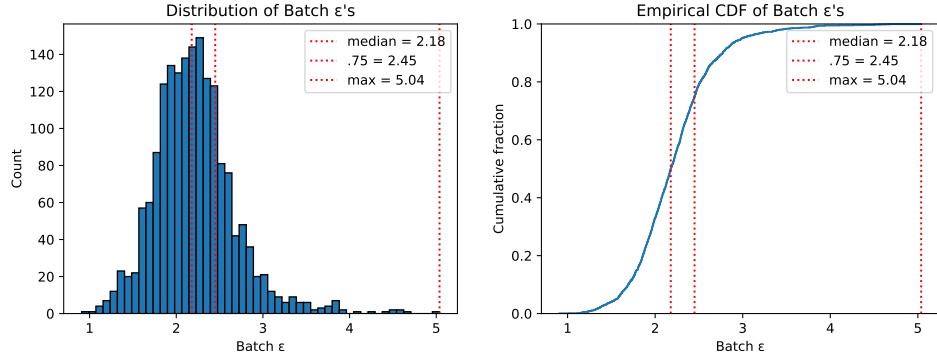

Figure 4: We plot the distribution of *per-batch $\varepsilon$ costs* of the median mechanism on NYT Topic. The maximum over all batches obtains $\varepsilon = 5.04$, which is the privacy guarantee we report in Table 3 via Theorem 2. *In this case, our accounting suffers particularly from the long right tail of per-batch privacy costs.*

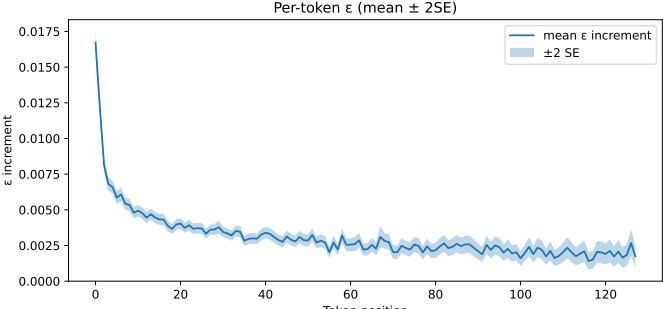

Figure 5: We plot the average *per-token $\varepsilon$ costs* of the median mechanism on AGNews ($\varepsilon = 2.40$). Consensus builds throughout generation, decreasing the privacy cost.

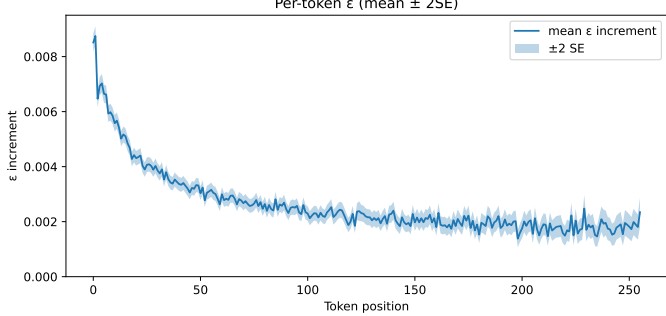

Figure 6: We plot the average *per-token $\varepsilon$ costs* of the median mechanism on Yelp ($\varepsilon = 2.21$).

## D.5 Empirical privacy tests

It is natural to consider the practical privacy ramifications of the relaxed privacy definition given in Definition 2. To investigate this, we supplement our theoretical analysis with empirical privacy tests.

We conduct two empirical privacy tests: (1) a **reconstruction test**, where we run our method on a dataset consisting of secret keys and measure the fraction of them emitted as synthetic data (comparing against a control: the fraction emitted belonging to an unseen, holdout set); and (2) a **privacy audit** on the median mechanism that yields an empirical lower bound of the epsilon. Both tests do not demonstrate evidence of privacy violations.

### D.5.1 Reconstruction test

**Setup.** We form a dataset of 50k unique 5-digit strings, each one randomly selected from `["00000", ... "99999"])`, creating entries that take the form `"My 5-digit key is 00000."` We randomly select half (25k) as source data input to our DP inference algorithm and reserve the other half (25k) as holdout data. *A gap in the number of exact matches in the synthetic data for source and holdout is evidence of memorization.*

We run the median algorithm with the exact same hyperparameters as in Table 3, and obtain an ex-post, data-dependent guarantee. To strengthen the reconstruction test, we prefill the generated output with the prefix `"My 5-digit key is "` and generate only the numbers afterward.

**Reconstruction rates are comparable to random data.** In Table 7, we see that source data is not more likely to be emitted than unseen holdout data, suggesting no evidence of memorization. Furthermore, notice that in this setup, drawing a random set of 5-digit numbers and treating that as the synthetic-data-under-test would yield source and holdout overlap rates of 25% which is comparable to these results. Finally, we also run the same test for unconditional $\varepsilon = 3$ mean and observe the same result as our data-dependent $\varepsilon = 2.29$ median.

| Method | Privacy $\varepsilon$ | Synthetic count | % Source overlap (count) | % Holdout overlap (count) |
|--------|-----------|-----------------|--------------------------|---------------------------|
| Median | 2.29* | 7685 | 25.5% (1958) | 25.8% (1979) |
| Mean | 3 | 8268 | 24.8% (2051) | 25.2% (2087) |

Table 7: Results of the reconstruction test. (*) denotes data-dependent $\varepsilon$. Both the mean run with unconditional $\varepsilon = 3$ and the median run with data-dependent $\varepsilon = 2.29$ demonstrate that generated synthetic data's overlap with input source data does not significantly differ from overlap with holdout data that is unseen by the method.

### D.5.2 Empirical audit of the median mechanism

To address concerns regarding possible gaps between the implemented method and the analyzed algorithm, we conducted an empirical privacy audit on the median method. We found no violations of stated $\varepsilon$ upper bounds.

**Data-dependent epsilon calculation is already empirical.** We remark that the nature of our data-dependent calculation of epsilon (Theorem 2) is highly similar to running an empirical privacy audit. Rather than the usual paradigm in unconditional DP of (1) writing down an algorithm, (2) proving an $\varepsilon$ guarantee; and (3) trusting the algorithm to be correctly implemented; in the ex-post data dependent regime, we (1) implement the algorithm without stating any privacy guarantees a priori; and then (2) directly measure the privacy cost by examining the sampling distributions produced by the algorithm (akin to how privacy audits are done for private prediction (Chadha et al., 2024). Hence an additional privacy audit does not introduce a qualitatively new safeguard in this setting, compared to unconditional DP.

The gap remaining in the $\varepsilon$ upper bound in our analysis is that it makes the worst-case assumption that the adversary can craft prefixes that would optimally shift the median: both coordinate-wise and for each time step, to ensure maximal possible divergence.

**Setup.** For our empirical audit, we generate text from a batch, and compute the empirical privacy loss for an adversary that can remove any single context from the batch up-front. This mirrors the realistic counterfactual for a user that wants to assess the change in probability of the outcome given that they removed their example. To compute the privacy loss for removing a given example, we simply compute the sampling distribution at all time steps with or without the single example, and compute the gap in log probability of the output sequence. Taking the maximum over all examples over the batch yields the result of our empirical audit.

**Results.** We audit the clustered median algorithm that reports an epsilon of 2.40 in Table 3. Recall also that our privacy analysis uses parallel composition, so the reported 2.40 refers to the batch with the highest privacy cost (see Figure 3 for the full distribution). In Table 8, we present the results of auditing one batch from each of the 4 label categories in AGNews. Indeed, we do not find any evidence that our theoretical upper bounds are violated.

| Label of batch | Data-dependent $\varepsilon$ (upper bound) | Empirical $\varepsilon$ (lower bound) |
|---|---|---|
| World | 1.41 | 0.47 |
| Sports | 1.96 | 0.66 |
| Business | 1.37 | 0.53 |
| Science/Technology | 1.21 | 0.57 |

Table 8: Comparison of theoretical upper bounds and empirical lower bounds for the data-dependent $\varepsilon$ for various data batches. No evidence of violating bounds from the analysis.

### D.6 COMPARISONS TO PRIVATE EVOLUTION

While we focus on improved methods for private inference, it is important to note that private evolution techniques (Lin et al., 2023; Xie et al., 2024) offer a different approach to generating private synthetic data without fine-tuning a model.

Private evolution methods like Aug-PE (Xie et al., 2024) are much more sensitive to prompt design, since the synthetic data is generated from the prompt alone. For example, when generating a synthetic version of the Yelp dataset, the prompt used by Xie et al. (2024) includes the rating and the category of the review to be generated. The rating is from 1 to 5, and the category represents whether the review corresponds to a restaurant or hotel, etc. Furthermore, seed data is generated with side knowledge of possible further subcategories of reviews, such as the type of restaurant and the food it serves. On the other hand, private inference methods use private data directly to generate synthetic data, and therefore are able to use a trivial prompt ("Generate an example like this: ...") that does not require side knowledge about the private data.

To show how much private evolution methods can depend on prompt quality, we applied Aug-PE to the Gemma 2 2b model and the YELP-POLARITY dataset, which contains 500K reviews and binary review labels. We included only the review label in the prompt, but not the category of the review. After 10 epochs, the synthetic text was mostly meaningless or not in English, and had very low downstream accuracy and MAUVE score.

## E EXPERIMENTAL DETAILS

### E.1 EVALUATION HYPERPARAMETERS

**MAUVE.** The absolute value of MAUVE scores can vary due to the precise implementation details, however the relative rankings it assigns to datasets is robust (Pillutla et al., 2021). We follow the original implementation[2] closely, as well as report all hyperparameters used: 768-dim Gecko embeddings (Lee et al., 2024), $n = 1000$ texts per set, $n/10 = 100$ clusters (as recommended), $k$-means iteration limit of 500, 5 $k$-means initializations, PCA target explained variance of 0.9, MAUVE scaling factor of 5, and 32 MAUVE divergence curve discretization points.

---

[2]See `https://github.com/krishnap25/mauve/blob/main/src/mauve/compute_mauve.py`

**BERT.** We compute BERT accuracy on a synthetic dataset by first partitioning it into a synthetic validation and synthetic train component, then running a hyperparameter sweep for BERT training, and finally selecting the checkpoint with best synthetic validation accuracy, and then finally reporting the accuracy of the selected checkpoint on real held-out data.

The fraction of validation data is 0.1. We fix a batch size of 200, and train for roughly 500 steps by setting `epochs = math.ceil((batch_size * steps)/train_set_size)`. We use Adam, and search over 5 learning rates [1e-6, 3e-6, 1e-5, 3e-5, 1e-4] $\times$ 2 settings for weight decay [0.0, 5e-4]. In each run, we employ early stopping: stopping after 4 epochs with no improvement in synthetic validation accuracy and returning the best checkpoint so far, in terms of synthetic validation accuracy.

### E.2 METHOD HYPERPARAMETERS

| Setting | Model | Examples $k$ | Batch size | Output tokens | Temp. | Clip $c$ |
|---|---|---|---|---|---|---|
| Baseline | IT | 1 | 64 | 1000 | 1.5 | 9 |
| Baseline++ | PT | 2 | 64 | 1000 | 1.5 | 9 |
| *+ non-private clustering* | PT | 2 | 64 | 1000 | 1.5 | 9 |

Table 9: Hyperparameters for Yelp100k at $\varepsilon = \infty$ results presented in Table 1. $k$ refers to the number of examples in each context; $c$ is the clipping parameter in Equation 3.

| Setting | $\varepsilon$ | Model | Examples $k$ | Batch size | Output tokens | Temp. | Clip $c$ |
|---|---|---|---|---|---|---|---|
| Mean Baseline | 10 | IT | 1 | 64 | 373/337/355 | 1.5 | 9 |
| Mean Baseline++ | 10 | PT | 2 | 64 | 373/337/355 | 1.5 | 9 |
| Mean Clustered | 9.9 | PT | 2 | 64 | 367/331/349 | 1.5 | 9 |
| Median Clustered | - | PT | 2 | 64 | 367/331/349 | 1.5 | 6 |
| Mean Baseline | 3 | IT | 1 | 256 | 733/642/686 | 1.5 | 9 |
| Mean Baseline++ | 3 | PT | 2 | 256 | 733/642/686 | 1.5 | 9 |
| Mean Clustered | 2.9 | PT | 2 | 256 | 689/604/645 | 1.5 | 9 |
| Median Clustered | - | PT | 2 | 256 | 689/604/645 | 1.5 | 6 |

Table 10: Hyperparameters for $\varepsilon = 3$ and $\varepsilon = 10$ results presented in Table 3. The same hyperparameters are used across all datasets; except per-batch *Output tokens* which depends on input dataset size to target the same $\varepsilon$; we report results for *AGNews/Yelp/NYT Topic* respectively. $k$ refers to the number of examples in each context; $c$ is the clipping parameter in Equation. 3.

### E.3 DATASETS AND MODELS

Table 11 summarizes all the datasets and models used our experiments.

### E.4 PROMPTS

PT and IT Gemma variants necessitate changes prompt changes. We use the same templates across all datasets. We stop generation when the model outputs its respective end token: "` ``` `" for PT, "`<end_of_turn>`" for IT.

For clarity of exposition, we show the prompts when we use two examples per context, but the same template is generalizes to $k$ prompts per context (including $k = 1$ used in our experiments). The

---

[3]`https://huggingface.co/datasets/fancyzhx/dbpedia_14`
[4]`https://huggingface.co/datasets/fancyzhx/ag_news`
[5]`https://huggingface.co/datasets/fancyzhx/yelp_polarity`
[6]`https://huggingface.co/datasets/dstefa/New_York_Times_Topics`
[7]`https://huggingface.co/google/gemma-2-2b-it`
[8]`https://huggingface.co/google/gemma-2-2b`
[9]`https://huggingface.co/google/bert_uncased_L-10_H-256_A-4`

| Dataset | $n_{train}$ | Description | Usage | Source |
|---|---|---|---|---|
| DBPedia | 560,000 | 14-category Wikipedia article topic | Public clusters | (Zhang et al., 2015)[3] |
| AGNews | 108,000 | 4-way news topic classification | Synthesis target | (Zhang et al., 2015)[4] |
| Yelp Polarity | 504,000 | 2-way review sentiment classification | Synthesis target | (Zhang et al., 2015)[5] |
| NYT Topics | 230,400 | 8-way news topic classification | Synthesis target | (Singh, 2021)[6] |

(a) Overview of datasets used. For synthesis targets, $n_{train}$ is 10% smaller than reported elsewhere as we split off that amount to use for validation.

| Model | Usage | Source |
|---|---|---|
| Gecko | Generation; embeddings for clustering | (Lee et al., 2024) |
| Gemma 2 2B IT | Generation; DP Inference | (Gemma Team, 2024)[7] |
| Gemma 2 2B PT | | (Gemma Team, 2024)[8] |
| BERT-Base 12/768 110M | Evaluation; downstream finetuning | (Turc et al., 2019)[9] |
| Gecko | Evaluation; embeddings for MAUVE | (Lee et al., 2024) |

(b) Overview of models used in experiments.

Table 11: Datasets and models used in our experiments. The Gecko embedding model is used for clustering, as well as for computing MAUVE. Gemma 2 2B IT/PT are used for synthetic data generation. We finetune BERT using synthetic data to evaluate how useful synthetic data is for improving accuracy on real data.

field `label` is in natural language (e.g. `Positive` or `Negative` for Yelp Polarity); and recall that batches are constructed so that that all examples in a batch share a label.

### E.4.1 PT PROMPT TEMPLATE

```
```
{label}
{example1}
```

```
{label}
{example2}
```

```
```

### E.4.2 IT PROMPT TEMPLATE

```
<start_of_turn>user
Here are texts with Label: {label}.

Text: {example1}

Text: {example2}

Please give another one. No formatting or explanations.<end_of_turn>
<start_of_turn>model
Text:
```

