# OpenReview forum: "Clustering Improves Differentially Private Inference"
_ICLR.cc/2026/Conference — Submitted to ICLR 2026_

### Official Review · Reviewer_dwQB · 2025-10-24

**Soundness:** 3
**Presentation:** 3
**Contribution:** 2
**Rating:** 6
**Confidence:** 3

**Summary:**

This paper studied the problem of private synthetic data generation and proposed an optimized version of private prediction based solution. Specifically, the authors identify that uniformly sampling ICL examples can lead to degraded utility due to heterogeneous distribution within the sample batch. To encounter this problem, the author proposed a DP clustering method by first calculating k cluster centers using public data, and then DP select the top-k' centers upon private data to avoid skewed distribution. The authors also proposed using median aggregation rather than mean aggregation to lower the sensitivity for clustering-based batching strategy. The experiment results shows improved utility and privacy cost.

**Strengths:**

- The paper is well-written and easy to follow.
- Applying DP for private synthetic data generation is an important direction.
- Extensive experiments on clustering and empirical private estimates have been conducted.

**Weaknesses:**

- The contribution of this paper is somewhat limited. The main utility improvement comes from the clustering-based batching for DP sampling. In [1], the authors have already discussed the choice of non-uniform batching, and assign prompts with the same label to the same batch. This work instead proposed to cluster based on the embedding similarity. I wonder if the authors could better clarify the difference.
- According to the ablation studies, the utility improvement mainly comes from using PT models and add more ICL samples. The clustering contributes marginal utility improvement according to Table 3 and Table 4.
- The selection of the cluster number $k$ is not clearly stated. Is it data dependent? Besides, does the $\epsilon$ in the DP top-$k'$ selection has significant impacts? That is,  is the top-$k'$ clusters consistently being selected? According to Figure 2, the top-100 public centers almost includes all the samples.

[1]: Private prediction for large-scale synthetic text generation. https://arxiv.org/pdf/2407.12108

**Questions:**

- The authors discussed the comparison against PE methods. I wonder if there will be similar improvements if PE methods refer to PT models rather than IT models.
- Regarding the explanation of why median aggregation yields a smaller sensitivity, the authors wrote ' if Z contains at least 3 identical vectors, then the local sensitivity of median(Z) is zero.'. I have two questions : 1) in this case, add/remove a sample causes no impact to the algorithm, which should somehow degrade the utility; 2) this example does not make sense to me. Since the batch is sampled from a cluster, which implies a similar embedding distribution, there should be few outliers in the batch. In this case, the sensitivity for mean aggregation (theoretically 2C/B) should be similar to that of median aggregation (2C).
- The refined understanding of privacy risk is promising. Could the authors elaborate on the criteria for private/public tokens and how the data-dependent accounting is designed?

---

> ### Author Response · Authors · 2025-11-21
> **Response to Reviewer dwQB**
>
> We thank the reviewer for their time and feedback. We found their comments to be insightful, and they helped us improve our paper’s clarity. We hope our rebuttal resolves the reviewer’s concerns.
>
> ---
>
> > Contributions (PT model, clustering and number of in-context examples)
>
> * **Changing from IT model to PT**: While this change seems trivial, we believe this is an important step toward more aligned synthetic data. Prior methods such as Private Evolution (PE) rely heavily on prompt engineering to generate high-quality data, and therefore they must utilize IT models. We believe our contribution is not changing the model from IT to PT. Rather it is designing an algorithm that does not rely on prompt engineering. Directly applying the PE method on the PT model would degrade performance, since the PT model is not promptable.
>
> * **Label batching VS clustering**: The goal behind label batching in [1] is to have consistent labels between the input and output so they can better evaluate the classification accuracy. However, after clustering (batching) based on the labels, the input is still very heterogeneous. Our contribution is to identify this heterogeneity and explore various ways to make the data more homogenous in a private manner.
>
> Furthermore, we share some additional results on the Yelp dataset, demonstrating **consistent improvement of clustering across all unclustered baselines** [$\epsilon$=$\infty$ for all settings]. The improvement from clustering is significant, consistent, and supported by all available evidence.
>
> |**Model**|**ICL examples**|**Cluster**|**Mauve**|**Accuracy**|
> |-|-|-|-|-|
> IT|1| 2|0.130 (0.009)|0.892 (0.015)|
> IT|1|200|0.222 (0.027)|0.921 (0.001)|
> IT|1|500|0.241 (0.036)|0.904 (0.017)|
> PT|1|2|0.385 (0.015)|0.918 (0.008)|
> PT|1|200|0.571 (0.016)|0.919 (0.002)|
> PT|1|500 |0.615 (0.056)|0.931 (0.005)|
> IT|2|2|0.203 (0.024)|0.891 (0.031)|
> IT|2|200|0.371 (0.019)|0.896 (0.021)|
> IT|2|500|0.352 (0.032)|0.888 (0.016)|
> **PT**|2|**2**|**0.460 (0.050)**|**0.898 (0.022)**|
> PT|2|200|0.663 (0.031)|0.898 (0.011)|
> **PT**|2|**500**|**0.650 (0.021)**|**0.912 (0.020)**|
>
> In our paper, we present improvements from clustering over the strongest possible baseline. A stronger baseline should be seen as a positive for the paper, as it demonstrates gains from clustering that are not subsumed by other improvements.
>
> > DP-clustering
>
> * **Number of K**: We thank the reviewer for pointing this out. We will add more clarification about this to the paper. In our algorithm, $K$ has two impacts. Larger $K$ means we have more granular, homogeneous and uniform clusters but fewer examples per cluster. While higher homogeneity is desired for better quality, having a cluster size smaller than batch size $\times$ the number of in-context examples means that we cannot use that cluster at all. In our setting, we selected $K$ such that we have enough number of clusters and number of datapoints in each cluster. We have also included an ablation study in Table 5 in D.2 (appendix).
>
> > PT model for PE method
>
> For a fair comparison in Section D.6 we used the same (PT) model and prompt as ours. However, since PE requires very fine-grained prompt and description of datapoints it fails to generate meaningful data when the prompt is general.
>
> > Add/remove a sample with no impact to the algorithm
>
> This is a hypothetical example meant to illustrate how the local sensitivity of the median can be much lower than the global sensitivity of the mean. We are not claiming that extreme cases like this are common in practice.
>
> > Sensitivity of mean vs median
>
> This is a very insightful question. This is correct, in most clusters because the outputs are similar the **actual** sensitivity of mean and median methods should also be similar. But, the mean privacy analysis is a worst-case scenario and does not care about the majority of cases. Since it does not take the homogeneity of input dataset into account, the mean method with approximate DP guarantee will give us much higher privacy cost compared to the median method with data-dependent DP analysis.
>
> ---
>
> We are very grateful for the reviewer's feedback. We hope that our rebuttal has addressed their concerns and they re-evaluate our work.

---

> > ### Comment · Reviewer_dwQB · 2025-11-27
> >
> > Thank the authors for the detailed response. Most of my concerns have been addressed, and I would like to maintain the positive score.

---

### Official Review · Reviewer_3wGe · 2025-10-29

**Soundness:** 3
**Presentation:** 3
**Contribution:** 2
**Rating:** 6
**Confidence:** 3

**Summary:**

This paper studies private synthetic data generation via LLM inference. Prior work follows a subsample-and-aggregate approach: split the private data uniformly at random into batches, query the LLM to get next-token logits for each record (non-privately), and then privately aggregate those logits to sample tokens. While this method can yield good downstream task accuracy, the synthetic text may be a poor representative of the original distribution because uniform batching mixes heterogeneous examples.

The authors propose batching similar examples together to improve representativeness and reduce privacy cost. They first compute public cluster centers using a non-private dataset, then privately rebalance by adding DP noise to cluster counts and keeping the top-$k$ centers to ensure that the clusters are not too small while spending a small privacy budget. Then, for each batch they compute logits for all seeds in the batch, apply clipping, and aggregate by the median rather than the mean. They analyze this with a data-dependent DP guarantee: when a batch’s logits are concentrated (small “median gaps”), the local sensitivity is smaller, which yields a lower $\epsilon$. Clustering makes batches more homogeneous, so the realized privacy loss can be lower than worst-case bounds.

Empirically, the method improves both representativeness metrics (e.g., MAUVE) and downstream accuracy compared to prior DP inference baselines. However, the overall privacy guarantee depends on the data and outputs.

**Strengths:**

- The paper is well written and easy to follow.

- Their proposed methods is simple but effective. It does a small change to prior DP-inference pipelines (replacing uniform batching and mean aggregation with cluster-informed batching and median aggregation) that is easy to adopt. The proposed method produces more representative synthetic data (higher MAUVE) and strong downstream accuracy in comparison with previous methods.

**Weaknesses:**

- The privacy guarantee is ex-post and data-dependent, which is weaker. For example, releasing the realized $\epsilon$ may leak information about the private data’s “median gaps.”

- MAUVE is lower on Yelp, which is farther from DBPedia, suggesting that public–private distributional closeness matters; performance may degrade on niche private datasets.

**Questions:**

Would applying PTR/smooth sensitivity keeps the advantage of the method, while also give a data-independent privacy guarantee?
How does the number of clusters affect the results?

---

> ### Author Response · Authors · 2025-11-21
> **Response to Reviewer 3wGe**
>
> We thank the reviewer for their time and feedback. We found their comments to be insightful, and they helped us improve our paper’s clarity. We hope our rebuttal resolves the reviewer’s concerns.
>
> ----
>
> > ex-post Privacy Guarantee and private data’s “median gaps”
>
> We thank the reviewer for pointing this out.
>
> 1. First we wanted to reemphasize that our paper has several contributions such as clustering, utilizing pretrained models (PT) and increasing the number of context examples. These modifications alone have helped to improve the final performance. All of these contributions still follow the original privacy definition. The only method that utilizes the ex-post data-dependent privacy analysis is the median algorithm.
>
> 2. We respectfully propose that our ex-post data-dependent guarantee is not “weaker”, but rather “more refined”. As we argue in Section 5.2, standard, data-independent guarantees are based on worst-case global sensitivity. This approach is overly conservative, paying the same high privacy cost for all the tokens. Our privacy analysis instead allows us to allocate the privacy budget only where it is needed. As shown in Figures 5 & 6 (Appendix D.4), this is empirically validated: the per-token privacy cost decreases during generation.
>
> 3. We agree with the reviewer about the median gap problem. That’s why we believe clustering and median methods go hand in hand. A good clustering algorithm ensures that the data within the batch is indeed similar and does not have a considerably large gap, then the median algorithm utilizes this fact and proposes a more refined privacy guarantee.
>
> > Dataset Dependency (Yelp/DBPedia)
>
> This is a correct and insightful observation. However, we respectfully disagree with the reviewer about the fact that this is a weakness. On the other hand, we believe this observation is the core motivation behind our paper.
>
> This observation shows that better clustering and unifying each batch improves the performance. As this is the case for the Yelp dataset. However, we are sure more complex public data can improve the results.
>
> We selected the DBPedia dataset as we believed Wikipedia dataset contains a variety of concepts. For niche private datasets our utility is completely dependent on our knowledge of the data. If there is no public dataset aligned with the private data, then we pay this cost by having lower utility.
>
> > PTR/smooth sensitivity
>
> We agree with the reviewer that a propose-test-release-style mechanism, or an analysis based on the smooth sensitivity framework, would be a very interesting direction for future work.
>
> > Number of clusters
>
> We studied this exact question, and the results are available in Appendix D.2, "Ablation on the number of clusters," (Table 5).
>
> ----
>
> We are very grateful for the reviewer's feedback. We hope that our rebuttal has addressed their concerns and they re-evaluate our work.

---

> > ### Comment · Reviewer_3wGe · 2025-11-27
> >
> > Thank you for your response. My concerns have been addressed.

---

### Official Review · Reviewer_Wa8q · 2025-10-31

**Soundness:** 2
**Presentation:** 3
**Contribution:** 3
**Rating:** 4
**Confidence:** 4

**Summary:**

Summary: This paper addresses the problem that random sampling leads to heterogeneous batches and degrades data quality when generating synthetic text via differentially private (DP) language model inference. It proposes an improved method combining clustering and median aggregation. Core contributions include: 1) Constructing homogeneous batches by pre-clustering sensitive data to solve the heterogeneity issue of random sampling; 2) Designing a median-based logit aggregation algorithm that leverages the similarity of prediction results under homogeneous data to reduce local sensitivity, achieving a data-dependent ex-post DP guarantee; 3) Validating on datasets such as AGNews and Yelp that the method significantly outperforms existing baselines in MAUVE (distributional similarity) and downstream task accuracy with lower privacy cost.

**Strengths:**

1.Novel use of clustering as a preprocessing step to improve DP inference quality; novel median-based aggregation with a tailored data-dependent ex-post privacy analysis.
2.The paper pairs formal analysis (Algorithm 1, Theorem 2 and proof sketch) with practical experiments and empirical privacy checks. Results are presented across multiple datasets and ε settings.
3.The paper has a well-structured hierarchy, with coherent logic from problem formulation, method design to experimental verification. Explanations of core concepts (such as local sensitivity, data-dependent ex-post DP) are concise and clear.
4.It solves the key problem of "imbalance between quantity and representativeness" in DP synthetic data, especially suitable for sensitive heterogeneous data scenarios such as medical records. It promotes the practicalization of DP inference technology.

**Weaknesses:**

1.It relies on public datasets (DBPedia) to construct cluster centers. If the distribution of public data differs significantly from that of sensitive data, cluster imbalance may still occur; DP clustering methods perform poorly on high-dimensional text embeddings.
2.There is a lack of further attack experiments (such as membership inference/extraction for a small number of anomalous samples).
3. It fails to conduct a fair comparison with the latest DP synthetic data methods (such as API-based private evolution technology) under the same prompt and model settings.
4. The sensitivity analysis of hyperparameters such as the number of clusters and embedding models is insufficient, and no general criteria for optimal parameter selection are clarified.
5. Although the data-dependent ex-post DP guarantee is more accurate, the privacy cost needs to be calculated after generation, making it impossible to determine the privacy budget in advance. This increases the complexity of practical deployment.

**Questions:**

1.How was the initial public k (e.g., 1000 → rebalanced to 60/80) chosen? Any sensitivity/selection rule?
2. How should multiple per-output data-dependent ε values be combined into a final released ε? Which composition theorem is applied in practice?
3. This method is not compared with some of the latest baselines: Aug-PE[1], DP-fusion[2], DP-SynRAG[3], INVISIBLEINK[4].
4. The privacy cost of median aggregation depends on generation results. In practical deployment, how to balance the needs of "ex-post calculation" and "real-time generation"?

[1] Differentially private synthetic data via foundation model APIs 2: Text
[2] DP-fusion: Token-level differentially private inference for large language models
[3] Differentially private synthetic text generation for retrieval-augmented generation (RAG)
[4] InvisibleInk: High-utility and low-cost text generation with differential privacy

---

> ### Author Response · Authors · 2025-11-17
>
> Thanks for the review! Requesting a quick clarification on:
> > 2.There is a lack of further attack experiments (such as membership inference/extraction for a small number of anomalous samples)
>
> Appendix D.5.1 reports an extraction attack on a random digit dataset, where we find that training set digits were reproduced equally often as unseen holdout digits. For the requested further attack experiments ("extraction for a small number of anomalous samples"), was the reviewer suggesting to see if the same results hold if we inserted the digit canaries into a normal dataset?
>
> Regarding membership inference: note that DP inference methods only produce synthetic data and not a model. So the naive membership inference test methodology will highly resemble an extraction test (e.g. report member iff input is reproduced in generated data), unless the reviewer has suggestions on that front.

---

> ### Author Response · Authors · 2025-11-22
> **Response to Reviewer Wa8q**
>
> We thank the reviewer for their time and feedback. We found their comments to be insightful, and they helped us improve our paper’s clarity. We hope our rebuttal resolves the reviewer’s concerns.
>
> ---
>
> > Public data for clustering and distribution shift from public to private data
>
> In theory, this can happen. However, our goal in this paper is different. We show when relevant public data is available, carefully combining that with clustering leads to more uniform batches and improved performance of DP synthetic data. For this exact reason, we selected the DBPedia dataset as we believed Wikipedia dataset contains a variety of concepts. If there is no public dataset aligned with the private data, we pay this cost by having lower utility, but this does not weaken the findings of our paper.
>
> > Membership inference attacks
>
> In fact, we have an entire section devoted to empirical privacy testing in the appendix (see Section D.5). Since our method does not learn a model, conventional membership inference attacks cannot be applied. The reconstruction attacks described in Section D.5.1 are the nearest approximation. We find no privacy violations as a result of our empirical privacy testing.
>
> > Initial number of public clusters and rebalancing
>
> We thank the reviewer for pointing this out. We will add more clarification about this to the paper.
>
> Both of these numbers are hyperparameters we can fine-tune based on the datasets. We have a few rules about how to select these parameters.
>
> * If we assume that the centroids of each public cluster represent an average topic of that cluster, too many clusters means that each center is going to be very specific (the extreme case is when each cluster has only one data point). On the other hand, selecting a small number of clusters may increase the risk of combining different topics. We used AgNews and V-measure to fine-tune this parameter, then used the same public centers for the rest of the datasets.
>
> * For the number of clusters after rebalancing, we had similar principles. In our algorithm, the final number of private clusters ($K$) has two effects. Larger $K$ means we have more granular, homogeneous and uniform clusters but fewer examples per cluster. While higher homogeneity is desired for better quality, having a cluster size smaller than batch size $\times$ the number of in-context examples means that we cannot use that cluster at all. In our setting, we selected $K$ such that we have enough number of clusters and number of datapoints in each cluster. We have also included an ablation study in Table 5 in D.2 (appendix).
>
> > How should multiple per-output data-dependent $\epsilon$ values be combined into a final released $\epsilon$? Which composition theorem is applied in practice?
>
> [Rogers et al](https://arxiv.org/pdf/2306.13824) proved composition theorems for ex-post differential privacy.
>
> > Comparison with other baseline
>
> We appreciate the reviewer for bringing this up, we will discuss each of these works below:
>
> * **Aug-PE**: We have indeed compared our algorithm with this baseline. We used the same model and prompt. However, since Aug-PE method heavily relies on prompt engineering and fine-grained labels per data points, it fails to generate meaningful text under generic prompt and PT model (for further discussion please check out Section D.6 in the Appendix).
>
> * **DP-fusion**: The goal of this paper is not to generate new synthetic data, the authors aim to privatize an existing document before releasing. Therefore, this method is not comparable with ours.
>
> * **DP-SynRAG**: We want to point out this paper has been published into arxiv in October 2025 (after ICLR submission deadline) so we could not have included any comparison with this method. Also, they are targeting generating user’s response in a RAG framework while the underlying DB is private which is different from our goal.
>
> * **INVISIBLEINK**: This is a concurrent work that also targets differentially synthetic data generation. But we want to clarify that all of their contributions (DClip and Top-k + sampling) are orthogonal to our contributions and these two methods can be applied simultaneously.
>
> We will have a discussion about these works in the appendix.
>
> > Balance the needs of "ex-post calculation" and "real-time generation" in practice
>
> The computational cost of calculating the data-dependent ex-post value of $\epsilon$ is extremely modest compared to the computational cost of generating the synthetic data, and does not impede the real-time generation of synthetic data in any practical sense.
>
> ---
> We are very grateful for the reviewer's feedback. We hope that our rebuttal has addressed their concerns and they re-evaluate our work.

---

### Official Review · Reviewer_X6y8 · 2025-10-31

**Soundness:** 2
**Presentation:** 2
**Contribution:** 2
**Rating:** 2
**Confidence:** 2

**Summary:**

The paper investigates the quality of synthetic data generated via differentially private (DP) inference methods, where multiple LLM outputs seeded by different users’ data are aggregated to ensure privacy. The authors argue that heterogeneity among seed examples in each batch can harm the representativeness of the synthetic data. To address this, they propose a privacy-preserving pre-clustering approach that groups similar examples before aggregation and introduce a median-based aggregation (instead of averaging). Their experiments show improved MAUVE and accuracy scores compared to considered baseline.

**Strengths:**

- The paper addresses a practical limitation of existing DP inference methods by considering the effect of data heterogeneity.

- Proposes a simple and intuitive modification (clustering and median-based aggregation) that can be easily integrated into existing frameworks.

**Weaknesses:**

- I find the argument in line 52 to be flawed. I think it incorrectly conflates privacy guarantees with data representativeness or quality. Differential privacy doesn’t require the aggregated response to be “representative” of the data; it only ensures that the presence or absence of any individual record (seed example) doesn’t significantly affect the output distribution.

- Table 3 is difficult to interpret because the reported ε values are not directly comparable. As far as I understand, the methods are based on different privacy formulations i.e data-dependent ex-post differential privacy versus approximate differential privacy, which provide distinct guarantees and should not be contrasted using the same ε metric.

**Questions:**

- The process for selecting top-k cluster centers and reassigning seeds is unclear [line 244]

- Would using median aggregation alone (without clustering) yield similar improvements?

---

> ### Author Response · Authors · 2025-11-21
> **Response to Reviewer X6y8**
>
> We thank the reviewer for their time and feedback. We found their comments to be insightful, and they helped us improve our paper’s clarity. We hope our rebuttal resolves the reviewer’s concerns.
>
> ----
> > Clarification on line 52
>
> We agree with the reviewer that the formal Differential Privacy definition does not require ‘representativeness’. This sentence refers to one of the challenges in applying DP in practice. Naively applying DP (specifically averaging) to heterogeneous data hurts performance (not the privacy guarantee), while the same algorithm with the same privacy guarantee can achieve considerably better performance if the underlying private data is more homogenous.
>
> This observation led us to look into ways that make the data more homogeneous (i.e., with clustering) that can result in better data quality with the same privacy guarantee.
>
> That said, we appreciate the reviewer’s comment for this point and we will revise the final manuscript to reflect this comment.
>
> > Comparison of $\epsilon$ values in Table 3
>
> We agree with the reviewer that the two $\epsilon$ values come from two different privacy definitions. The intention of Table 3 is to show:
>
> 1- Baseline++ and Mean Clustered (methods we have proposed in this paper) achieve better performance (in terms of accuracy and MAUVE score) compared to prior work (Mean Baseline) on all the three datasets.
>
> 2- Median Clustered also outperforms prior work on all three datasets, and is amenable to a new and tighter privacy analysis that better reflects the privacy cost of releasing synthetic data.
>
> The asterisk (*) and the caption (Line 410) flag the distinction between the privacy definitions. For further clarity, we will add a paragraph to the main manuscript explicitly mentioning the fact that these two $\epsilon$s come from different privacy analyses.
>
> > Selecting top-k cluster centers and reassigning seeds
>
> We explained public centers in lines 262- 269, and public centers with rebalancing in line 270 - 276.
>
> We will explain our algorithm here and add extra clarification to the appendix.
>
> 1- We obtain $k$ cluster centers (e,g, $k=500$) by running k-means on a *public* dataset such as DBPedia (no privacy cost).
>
> 2- We assign all our *private* data to these 500 public centers.
>
> 3- We compute a *differentially private noisy count* of the data points assigned to each cluster. This step incurs a small, one-time privacy cost ($\epsilon \approx 0.1$, which we report in Table 3, e.g., '$9.90+0.1$').
>
> 4- Finally, we select the $k'$ centers (e.g., $k'=100$) with the *highest noisy counts* (the 'top-k'' centers) and *discard* the rest.
>
> 5- We then *re-assign all* private seeds *only* to these 100 final, balanced centers.
>
> This rebalancing is what produces the much more balanced and usable clusters shown in Figure 2(d).
>
> > Median aggregation (without clustering)
>
>  We thank the reviewer for their insightful question. Below we have added the performance of the median algorithm on the AgNews dataset without clustering in the setting of Table 3.
>
> Median without clustering performs similarly to unclustered mean, but yields a smaller ex-post data-dependent DP guarantee.
>
> **Method** | **Privacy epsilon** | **Mauve** | **Accuracy** |
>  | :- | :- | :- | :- |
> Unclustered Mean     | 10   |  .633 (.022) | .851 (.015) |
> Unclustered Median  |2.42 | .643 (.002) | .859 (.009)|
> Unclustered Mean     |3      | .622 (.024) | .833 (.006)|
> Unclustered Median |1.24 | .636 (.024) | .829 (.005)|
>
>
> We would like to thank the reviewer once again and we hope that our responses and revision have addressed your concerns.
>
> To summarize, we commit to the following key revisions:
>
> 1. Revise the introduction line 52 to explain that representativeness is required for better utility.
>
> 2. Add a paragraph explaining the difference between the $\epsilon$s in Table 3.
>
> 3. Add further explanation for the 'Public centers with rebalancing' algorithm.
>
> 4. Add an experiment for the Median method without any clustering.
>
> -----
> We hope the reviewer will re-evaluate our work in light of these clarifications.

---

> > ### Comment · Reviewer_X6y8 · 2025-11-28
> > **Response**
> >
> > I'd like to thank the authors for their detailed responses and for addressing the concerns raised in my initial review. While I remain somewhat unconvinced about the method’s reliance on a large, high-quality public dataset for clustering and I continue to wonder how well this approach would generalize in domains where such public data is sparse, I acknowledge that this limitation is not inherent to the core technical contribution. Given the paper’s assumptions, the proposed method is reasonable and potentially impactful, and I am therefore comfortable raising my score. I wish the authors the best of luck.

---

### Author Response · Authors · 2025-12-03
**Summary of the review response**

This comment is intended to summarize the reviews, rebuttals and discussion in order to help the AC arrive at their decision.

## **Final ratings**

Three out of the four reviewers responded to our rebuttal.

* reviewer X6y8 (original rating: 2) **raised their rating, but the change was reverted due to the OpenReview data leak, and we do not know what the new score was**. The reviewer said: *“While I remain somewhat unconvinced about the method’s reliance on a large, high-quality public dataset for clustering and I continue to wonder how well this approach would generalize in domains where such public data is sparse, I acknowledge that this limitation is not inherent to the core technical contribution. Given the paper’s assumptions, the proposed method is reasonable and potentially impactful, and I am therefore comfortable raising my score. I wish the authors the best of luck.”*
* reviewer 3wGe (original rating: 6) acknowledged that their concerns have been addressed.
* reviewer dwQB (original rating: 6) acknowledged that most of their concerns have been addressed.


## **Strengths of our paper mentioned by the reviewers**

* The paper is easy to follow and has a well-structured hierarchy. [reviewer Wa8q, 3wGe, dwQB]
* Important observation regarding the limitation of existing DP inference methods: the effect of data heterogeneity. [reviewer X6y8, Wa8q]
* Intuitive and practical solution that can be integrated into the existing frameworks. [reviewer X6y8, 3wGe]
* Novel approach to improve DP inference quality: 1) clustering, 2) novel median-based aggregation with a tailored data-dependent ex-post privacy analysis. [reviewer Wa8q, 3wGe]
* The paper pairs formal analysis (Algorithm 1, Theorem 2 and proof sketch) with practical experiments and empirical privacy checks. [reviewer Wa8q]
* Results are presented across multiple datasets and ε settings. [reviewer Wa8q, dwQB]
* The proposed method produces more representative synthetic data (higher MAUVE) and strong downstream accuracy in comparison with previous methods. [reviewer 3wGe]

## **Our responses to weaknesses and questions raised by the reviewers**

* **Learning clusters on public data**: In this paper we show that better clustering and unifying each batch improves the performance of DP synthetic data generation. This means more complex public data can lead to better results.  Also, for niche/unique private datasets our utility is completely dependent on our knowledge of the data. If there is no public dataset aligned with the private data, then we pay this cost by having lower utility.

* **Comparison with other baselines**: In appendix (Section D.6) we have the comparison with Aug-PE method using the same model and prompt as ours. Other baselines requested by the reviewers either are orthogonal to our method or target DP data generation for other settings such as RAG or document privatisation.

* **Experiments requested by the reviewers that are already in the paper**:
     * Membership inference attacks: Section D.5.1
     * Number of clusters: Section D.2

## **Modifications that we agreed to implement**

* Revise the introduction line 52 to explain that representativeness is required for better utility.
* Add a paragraph explaining the difference between the $\epsilon$s in Table 3.
* Add further explanation for the 'Public centers with rebalancing' algorithm.
* Add an experiment for the Median method without any clustering.
* Add explanation for the parameter selection in the clustering step.

---

### Meta-Review · Area_Chair_osto · 2026-01-06

**Summary:**

This paper focuses on inference based data synthesis under DP consideration. A clustering based method is introduced, which produces high-quality synthetic data at lower privacy cost compared with existing works.

**Reviewer Concerns:**

- The method relies on public datasets to construct cluster centers, which may lead to cluster imbalance if the public and private data distributions differ significantly.
- The privacy guarantee is ex-post and data-dependent, which is inherently weaker, and the privacy cost can only be computed after data generation.
- Sensitivity analysis of key hyperparameters (e.g., number of clusters) is insufficient, and no general guidelines for selecting optimal parameters are provided.
- The paper does not provide a fair comparison with recent DP synthetic data generation methods.

While the authors address some of the concerns partially, the first two concerns seem critical and require more careful investigation.

**Reviewer Scores:**

The original scores were 2/4/6/6. The reviewer with score 2 indicated that they would increase the score during rebuttal, and the reviewers with score 6 indicated that they would maintain their score. For the reviewer with score 4, they would likely maintain their score as well.

---

### Decision · Program_Chairs · 2026-01-26

Reject